The impact of chemerin or chemokine-like receptor 1 loss on the mouse gut microbiome

Dranse Helen J. 1 2
Zheng Ashlee 3
Comeau André M. 1 4
Langille Morgan G.I. 1 4
Zabel Brian A. 3
Sinal Christopher J. csinal@dal.ca 1
1 Department of Pharmacology, Dalhousie University , Halifax , Canada
2 Toronto General Hospital Research Institute, University Health Network , Toronto , Canada
3 Palo Alto Institute for Research and Education, Veterans Affairs Palo Alto Health Care System , Palo Alto , United States of America
4 Integrated Microbiome Resource, Dalhousie University , Halifax , Nova Scotia , Canada
Aziz Ramy
Electronic publication date: 2018 Sep 12
Publication date: 2018
Volume: 6
Electronic Location ID: e5494
Received 2018 Mar 16; Accepted 2018 Jul 30
Copyright: ©2018 Dranse et al.
Copyright year: 2018
Copyright holder: Dranse et al.
License: This is an open access article distributed under the terms of the Creative Commons Attribution License, which permits unrestricted use, distribution, reproduction and adaptation in any medium and for any purpose provided that it is properly attributed. For attribution, the original author(s), title, publication source (PeerJ) and either DOI or URL of the article must be cited.
License URL: https://creativecommons.org/licenses/by/4.0/

Keywords: Microbiome, Chemerin, Chemokine-like receptor 1, Adipokine, Mouse, Adipose

Funding: Canadian Institutes of Health Research (CJS) United States National Institutes of Health R01AI-079320 This work was supported by a grant from the Canadian Institutes of Health Research (CJS) and United States National Institutes of Health grant R01AI-079320 (BAZ). The funders had no role in study design, data collection and analysis, decision to publish, or preparation of the manuscript.

==============================
Chemerin is an adipocyte derived signalling molecule (adipokine) that serves as a ligand activator of Chemokine-like receptor 1(CMKLR1). Chemerin/CMKLR1 signalling is well established to regulate fundamental processes in metabolism and inflammation. The composition and function of gut microbiota has also been shown to impact the development of metabolic and inflammatory diseases such as obesity, diabetes and inflammatory bowel disease. In this study, we assessed the microbiome composition of fecal samples isolated from wildtype, chemerin, or CMKLR1 knockout mice using Illumina-based sequencing. Moreover, the knockout mice and respective wildtype mice used in this study were housed at different universities allowing us to compare facility-dependent effects on microbiome composition. While there was no difference in alpha diversity within samples when compared by either facility or genotype, we observed a dramatic difference in the presence and abundance of numerous taxa between facilities. There were minor differences in bacterial abundance between wildtype and chemerin knockout mice, but significantly more differences in taxa abundance between wildtype and CMKLR1 knockout mice. Specifically, CMKLR1 knockout mice exhibited decreased abundance of Akkermansia and Prevotella, which correlated with body weight in CMKLR1 knockout, but not wildtype mice. This is the first study to investigate a linkage between chemerin/CMKLR1 signaling and microbiome composition. The results of our study suggest that chemerin/CMKLR1 signaling influences metabolic processes through effects on the gut microbiome. Furthermore, the dramatic difference in microbiome composition between facilities might contribute to discrepancies in the metabolic phenotype of CMKLR1 knockout mice reported by independent groups. Considered altogether, these findings establish a foundation for future studies to investigate the relationship between chemerin signaling and the gut microbiome on the development and progression of metabolic and inflammatory disease.

Introduction

The human body is host to a vast number of microbes that include bacteria, fungi, protozoan cells, and viruses, which live in a symbiotic manner encompassing commensal, mutual, and sometimes parasitic relationships, and are collectively termed the microbiota. The healthy human microbiota is comprised of approximately 1013 microorganisms that colonize the oral and nasal cavities, the skin surface, and the gastrointestinal (GI) and urogenital tracts (Barlow, Yu & Mathur, 2015; Sender, Fuchs & Milo, 2016; Whitman, Coleman & Wiebe, 1998). Together, the microbiota encode 150 times more genes than the human genome (Qin et al., 2010) and perform a number of functions that are important to human metabolism, energy homeostasis, neuro-hormonal function, and development of the immune system (Barlow, Yu & Mathur, 2015). In return, the microbiota benefit from the protective and nutrient-rich environment of the host. The composition of the gut microbial community co-develops with the host and is strongly influenced by several genetic and environmental factors. These include mode of birth, genotype, age, diet, antibiotic treatment, and exposure to factors such as pathogens and chemicals (Barlow, Yu & Mathur, 2015; Cox & Blaser, 2015; Marchesi et al., 2015). Disruption in the normal balance of gut microbial populations, or dysbiosis, results in profound changes in both the activity and function of intestinal microbiota. Increasing evidence from both animal models and human observational studies suggest that gut microbial dysbiosis is associated with a wide range of pathological conditions. These include obesity, diabetes, inflammatory bowel disease (IBD), liver disease, cancer, allergy and autoimMune diseases (Barlow, Yu & Mathur, 2015; Marchesi et al., 2015; Zhang et al., 2015).

Chemerin is a potent chemoattractant and adipokine that has been shown to play important roles in both metabolic and inflammatory diseases (for review, see Rourke, Dranse & Sinal, 2013; Zabel et al., 2014). Clinical studies have demonstrated that chemerin levels are positively associated with BMI and deleterious changes in glucose, lipid, and cytokine homeostasis (for review, see Rourke, Dranse & Sinal, 2013). Consistent with this, animal models have demonstrated that chemerin signaling through both chemokine-like receptor 1 (CMKLR1) and G protein coupled receptor 1 (GPR1) influence adiposity and glucose homeostasis (Ernst et al., 2012; Gruben et al., 2014; Rouger et al., 2013; Rourke et al., 2014; Takahashi et al., 2011; Wargent et al., 2015). Additionally, studies investigating the role of chemerin signaling in IBD pathogenesis have demonstrated that local chemerin expression, secretion, and activation increase in the colon and are positively associated with severity of inflammation (Dranse et al., 2015; Lin et al., 2014; Weigert et al., 2010). Together, this data suggests that chemerin serves as a link between obesity, inflammation, and other disorders that have previously been associated with changes in microbiome composition and activity.

In addition to these roles in metabolism and inflammation, two chemerin isoforms (chemerin-157 and chemerin-125) have been shown to exhibit potent antimicrobial activity against the growth of Escherichia coli and Klebsiella pneumoniae (Kulig et al., 2011). More recently, chemerin was demonstrated to act as an anti-microbial agent in human skin and provide protection against E. coli, Staphylococcus aureus, P. aeruginosa, and Candida albicans growth, which was directly mediated through an increase in bacterial lysis (Banas et al., 2013). In addition, Staphopain B, a S. aureus-derived cysteine protease, has been shown to act as a potent activator of chemerin (Kulig et al., 2007). This suggests that in addition to chemerin modulating microbial growth, the microbiota may also exert an influence on chemerin bioactivity. Based on these overlaps between chemerin signaling, metabolic and inflammatory disease, and known interactions with microbiota, we hypothesized that chemerin plays a role in regulating gut microbiome composition. The objective of this study was to investigate the diversity and relative abundance of microbiota in the lower GI tract in the presence or absence of chemerin/CMKLR1 signaling. To do this, we performed Illumina-based sequencing of the hypervariable V6-8 region of the bacterial 16s ribosomal (rRNA) gene on DNA extracted from stool samples obtained from healthy wildtype, chemerin knockout (KO), and CMKLR1 KO mice and examined changes in taxa abundance between genotypes.

Materials and Methods

Animals

Wildtype (WT) C57Bl/6 mice (Christopher Sinal (CS) colony) and CMKLR1 knockout (KO) mice were maintained in the Carlton Animal Care Facility at Dalhousie University (Halifax, NS, Canada). C57Bl/6 mice (CS) were obtained from the Jackson Laboratory (Bar Harbor, ME). CMKLR1 KO mice were originally created by Deltagen and were fully backcrossed onto the C57Bl/6 (CS) background as previously described (Ernst et al., 2012; Graham et al., 2009). WT C57Bl/6 mice (Brian Zabel (BZ) colony) were obtained from Jackson laboratory. Chemerin KO mice raised on a C57Bl/6 background were maintained in Veterinary Medical Unit at the Veterans Affairs Palo Alto Health Care Systems (VAPAHCS, Palo Alto, CA, USA). Animals were housed in micro-isolator cages under specific pathogen free conditions (Helicobacter-, norovirus-, and parvovirus-free) and all mice had free access to food and water in home cages. Mice at Dalhousie University received Prolab RMH 3000 (LabDiet, St. Louis, MO, USA) and mice at Stanford University received Teklad 18% protein rodent diet (Harlan Laboratories, Madison, WI, USA). All WT and KO animals were generated from WT x WT or KO x KO breedings. At weaning (approximately 3 weeks of age), 2 WT and 2 KO female mice were co-housed in a single cage to remove any possible differences in microbiome resulting from “cage effects”. Female mice were used to comply with animal protocol relating to the co-housing of mice and aggression. In total, 18 WT (CS), 18 CMKLR1 KO, 18 WT (BZ) and 18 chemerin KO mice were used in the study. Animal protocols were approved by Dalhousie University Committee on Laboratory animals (14-064) in accordance with the Canadian Council on Animal Care guidelines and the Institutional Animal Use and Care Committee at the Veterans Affairs Palo Alto Health Care System.

Genotyping

Ear punches were collected from WT (CS) and CMKLR1 KO animals and digested in proteinase K buffer (0.1 M Tris pH 8.0, 5 mM EDTA, 0.2% SDS, 0.2 M NaCl, and 100 µg/mL proteinase K) at 55 °C for 120 min. DNA was isolated with isopropanol precipitation, washed in 70% ethanol, and resuspended in distilled water. Samples were genotyped using Taq polymerase (Invitrogen, Burlington, ON) with primers specific for the CMKLR1 locus (Primer 1: TACAGCTTGGTGTGCTTCCTCGGTC; Primer 2: TGATCTTGCACATGGCCTTCC; Primer 3: GGGTGGGATTAGATAAATGCCTGCTCT). Each PCR consisted of 30 cycles of 95 °C for 30 s, 60 °C for 30 s, and 72 °C for 30 s with standard PCR reagent conditions (200 mM Tris-HCl pH 8.4, 500 mM KCl, 2.5 mM MgCl2, 0.1 mM dNTPs, and 0.2 µM each primer). Products were visualized by ethidium bromide staining after electrophoresis on a 2.5% agarose gel using a DyNA Light UV transilluminator (Mandel, Guelph, ON), ESAS 290 electrophoresis analysis system (Kodak, Rochester, NY, USA), and 1D image analysis software (Kodak, Rochester, NY, USA). Genotyping of WT (BZ) and chemerin KO mice was performed as previously described (Banas et al., 2015).

Fecal sample collection

Following 3 weeks of co-housing (at approximately 6 weeks of age), fecal samples were collected from each mouse. Mice were placed into an empty sterile cage and left until stool samples were produced (maximum 10 min). 3–4 pellets were collected per animal (approximately 100 mg dry weight per collection period). Samples were transferred with sterile forceps to a clean microcentrifuge tube and frozen immediately at −80 °C until DNA isolation was performed. Fecal samples from WT (BZ) and chemerin KO mice were shipped on dry ice to Dalhousie University. Fecal samples from WT (CS) and CMKLR1 KO mice were also collected at 8 weeks of age. In total, 108 samples were collected. In addition, WT (CS) and CMKLR1 KO mice were weighed using a bench-top balance (Mettler Toledo, Mississauga, ON) at the time of stool collection.

DNA isolation, library preparation, and 16S sequencing

Genomic DNA was isolated from fecal samples using the PowerFecal DNA Isolation Kit (MO BIO Laboratories, Carlsbad, CA, USA). Approximately 2–3 pellets of stool from each mouse were processed in batches of 24 over 5 consecutive days as per manufacturer’s instructions. Isolated DNA was eluted in 100 µL of Solution C6 and stored at −80 °C. 16S ribosomal RNA gene fragments were amplified and sequenced as previously described (Comeau, Douglas & Langille, 2017). Briefly, duplicate dilutions of each DNA sample were amplified using a high-fidelity polymerase and full fusion primers (Illumina adapters, indices and specific regions) targeting the V6–V8 region of the 16S rRNA gene. The duplicate PCR products (438 bp of usable sequence) from each sample were pooled, verified via high-throughput gels, then purified and normalized as previously described (Comeau, Douglas & Langille, 2017). The final pooled library was concentrated using the DNA Clean & Concentrator-5 kit (Zymo Research, Irvine, CA, USA) and DNA was quantified using a Qubit fluorimeter (Invitrogen, Burlington, ON). The library was loaded into a MiSeq Sequencer (Illumina, San Diego, CA, USA) at 20 pM with a 5% PhiX spike-in and sequenced using a 600 cycle v3 kit (300+300 bp) according to the standard protocol recommended by the manufacturer.

Data analysis

Data analysis was performed according to the workflow outlined in Microbiome Helper (Comeau, Douglas & Langille, 2017) using the indicated programs (Andrews, 2010; Caporaso et al., 2010; Kopylova, Noe & Touzet, 2012; Langille, 2015; Parks et al., 2014; Zhang et al., 2014). Overall quality assessment of the sequencing run was performed on raw FASTQ files using FASTQC (Andrews, 2010). Paired-end reads were stitched together to produce the full amplicon sequence using PEAR (Zhang et al., 2014). Reads with an ‘N’, a length of less than 400 bp, or a quality score of less than 30 over 90% of the bases were removed from further analysis.

Taxonomic analysis and visualization

Sequencing reads were clustered into operational taxonomic units (OTU) with a 97% identity threshold using QIIME (Caporaso et al., 2010). An open-reference OTU picking protocol (Rideout et al., 2014) was performed using GreenGenes as a reference database of known sequences (DeSantis et al., 2006). All samples were normalized to a sequence depth of 15,000. All further analysis was first performed for all 108 samples combined, and then individually for all samples within a particular facility using QIIME (Caporaso et al., 2010 and references within). To visualize diversity within samples, alpha rarefaction plots were generated using a minimum of 1,000 sequences and a maximum of 15,000 sequences per sample over 15 steps. To examine differences between samples, both unweighted and weighted UniFrac beta diversity plots were generated using principal coordinate analysis (PCoA). Additional analysis of Akkermansia and Prevotella abundances (sums of observed sequences within each genus) was performed by dividing the samples into groups (high/low abundance) using the mean abundance of each genus as a threshold.

Statistical analysis

Total weight and weight gain data was graphed using GraphPad Prism (version 5.0b, La Jolla, CA, USA) and assessed using an unpaired Student’s t-test. Data (mean ±  standard deviation) for alpha rarefaction plots generated through the pipeline described above was graphed using GraphPad Prism. STAMP (Parks et al., 2014) was used to determine statistically significant differences within taxonomic ranks between samples. Two-sample comparisons were performed using a two-sided Welch’s t-test. Abundance values reported are the minimum, maximum, median, average number, and standard deviation of the number of sequences observed for a particular taxon within a single sample. Box plots of bacterial abundance were created using GraphPad Prism using the Tukey method to plot whiskers and outliers. The line and cross represent the median and mean, respectively, and outliers are represented as dots. Correlations between total weight and abundance values were determined using PAST (Hammer, Harper & Ryan, 2001) and significance was calculated using the Pearson’s r correlation test or the Spearman rank-order correlation test when data was not normally distributed (as determined by the Shapiro–Wilk normality test). Abundance values were represented as log(abundance) and abundance values of 0 were excluded in the analysis (one sample in the g_Akkermansia KO comparison). The significance of clustering within PCoA plots was statistically tested using an Adonis, a nonparametric multivariate method, within the QIIME package using the default 999 permutations. For all data, significance was reported as p < 0.05 and when appropriate was corrected for multiple testing using Benjamini–Hochberg false-discovery rate (FDR).

Results

To investigate the impact of a loss of chemerin or CMKLR1 expression on microbiome composition, 16S sequencing was performed on DNA isolated from fecal samples collected from female WT (BZ and CS colonies), chemerin KO, and CMKLR1 KO mice. There were no apparent differences between WT and KO mice in terms of general health and wellness. Overall, sequences were clustered into a total of 27,323 operational taxonomic units (OTUs) at a 3% identity threshold among all samples.

Species richness is similar between facilities and genotypes

We first examined alpha-diversity using the number of observed OTUs for wildtype (CS and BZ), CMKLR1 KO, and chemerin KO mice (Fig. 1). Overall there was no significant difference in alpha-diversity between both facility and genotype. However, there was a trend for slightly higher diversity in samples collected from VAPAHCS (WT (BZ): 1,467 ± 162; Chemerin KO: 1,517 ± 159) compared to Dalhousie University (WT (CS): 1,313 ± 157; CMKLR1 KO: 1,315 ± 201) at a depth of 15,000 reads.

Figure 1 Species richness is similar across both facility and genotype.

Alpha rarefaction plot demonstrating the number of observed OTUs (mean ± standard deviation) at the indicated number of 16S sequences sampled per group (minimum 1,000, maximum 15,000 sequences).

Mice housed in the two different facilities have dramatically altered microbiome profiles

We next investigated differences in the taxonomic composition of the microbiome between the four groups of mice. Principal coordinate analysis (PCoA) using both unweighted and weighted (Fig. 2) UniFrac demonstrated significant separation by facility (Adonis: p < 0.001). Principal coordinate 1 (PC1, percent variation explained: 13.69%) of the unweighted UniFrac distinctly separated mice between the two facilities indicating substantial differences in the presence/absence of OTUs between facilities. However, the weighted UniFrac explained a larger amount of variation between facilities (∼63%) after the introduction of PC2 (15.92%) and PC3 (15.16%). This demonstrates that a significant proportion of the difference in microbiome composition between facilities is related to the abundance of overlapping OTUs. Importantly, there was no observable separation between genotypes at either facility using both the unweighted and weighted UniFrac analysis when all four groups of mice were considered together.

Figure 2 Microbiome diversity and abundance is dramatically different between mice housed at VAPAHCS and Dalhousie University, but not between genotypes within a facility.

(A) Principal coordinate analysis (PCoA) of 16S sequences sampled from all four groups of mice using both unweighted and weighted UniFrac. Each point represents a different sample. (B) Changes in the abundance of different bacterial genera between wildtype mice housed at Dalhousie or VAPAHCS. Genera that composed more than 1% of sequences sampled and had significant differences in abundance between the two facilities are shown.

To further explore differences in microbiome composition between facilities, we compared the abundance of different OTUs in fecal samples isolated from WT mice at Dalhousie University (CS) and VAPAHCS (BZ). All significant differences between the two facilities at each taxonomic level are highlighted in Table 1. There were several differences at the phylum level, including increased abundance of Cyanobacteria and Verrucomicrobia, and decreased abundance of Actinobacteria, in CS mice compared to BZ. Analysis of the abundance table revealed that significant differences at a single genus level were responsible for most differences at higher taxonomic ranks. For example, the difference in phylum Verrucomicrobia was entirely due to increased abundance of Akkermansia in CS mice, and Bifidobacterium comprised most differences in the Actinobacteria phylum between facilities. As a result, we focused our analysis on differences at the genus level in order to gain the highest resolution when identifying microbial differences between Dalhousie University and VAPAHCS. There were changes in 22 different identified genera as well as 13 groups that were unclassified at the genus level, but associated with known families (Table 1). Many of these taxa comprise a very minor proportion of the total microbiome community. For instance, one Fluviicola sequence was identified in samples from CS mice but not in samples from BZ mice. Therefore, only changes in which the abundance of each genus was greater than 1% of total sequences sampled from at least one facility are shown in Fig. 2B. The abundance of Akkermansia, Desulfovibrio, and Ruminococcus was significantly higher in CS vs. BZ mice, while the abundance of Bacteroides, Bifidobacterium, Odoribacter, Parabacteroides, [Prevotella], and unclassified bacteria of the Clostridiaceae family was significantly lower in CS vs. BZ mice. Notably, CS mice had significant amounts of Akkermansia (∼3% of the total microbiome) whereas Akkermansia was nearly undetectable in BZ mice. In contrast, Bacteroides, Bifidobacterium, Odoribacter, [Prevotella], and unclassified members of the Clostridiaceae family were prevalent in BZ mice, compromising ∼8.5% of the gut microbiome, but these microorganisms were detected at extremely low abundance levels in CS mice. Altogether, these results confirm our earlier findings that differences in the microbiome composition between the two facilities is due to both the presence/absence of certain bacteria as well as the proportion of overlapping OTUs.

Table 1 Significant differences in the abundance of bacteria between wildtype mice housed at VAPAHCS (BZ) and Dalhousie University (CS).

Statistical analysis of 16S sequences isolated from the stool of wildtype mice housed in the two different facilities was performed. Significant differences at each taxonomic rank are indicated. Abundance values are shown for both groups of WT mice and represent the number of sequences identified out of a total of 15,000 sequences analyzed per sample. “Unclassified” denotes OTUs that do not have a specific classification but are known to be within the level specified. For simplicity, unclassified ranks are removed from further levels in the table. Taxonomic ranks are represented as k, kingdom; p, phylum; c, class; o, order; f, family; g, genus; and s, species. The minimum (min), maximum (max), median (med), and mean number of observed sequences in a single sample are shown. SD, standard deviation.

	Wildtype (CS)	Wildtype (BZ)		
Feature	Min	Max	Med	Mean	SD	Min	Max	Med	Mean	SD	p-value	
Phylum:												
p__Actinobacteria	1	29	10	10.8	6.9	27	139	44	58.7	36.0	2.87e−05	
p__Cyanobacteria	1	59	15.5	20.9	17.9	0	8	3.5	3.2	2.4	1.05e−06	
p__Verrucomicrobia	10	4,027	195	483	747	0	3	0	0.6	0.9	4.39e−04	
Unclassified k__Bacteria	0	40	2	5.0	7.9	0	1	0	0.1	0.3	7.20e−04	
Class:												
c__4C0d-2	1	59	14.5	20.6	17.9	0	8	3	2.9	2.3	1.08e−06	
c__Actinobacteria	0	2	0	0.1	0.4	11	122	35	47.3	33.8	1.64e−05	
c__Alphaproteobacteria	2	32	14.5	14.3	8.1	1	31	6.5	8.6	7.8	0.017	
c__Chloroplast	0	2	0	0.2	0.5	0	0	0	0	0	0.033	
c__Erysipelotrichi	1	61	12.5	16.7	14.9	10	214	80	86.2	56.9	7.50e−05	
c__Flavobacteriia	0	27	2.5	4.6	6.4	0	2	0	0.3	0.5	2.96e−04	
c__Gammaproteobacteria	0	10	1	2.3	2.9	0	3	0	0.5	0.8	1.49e−03	
c__Verrucomicrobiae	10	4,027	195	484	747	0	3	0	0.7	0.9	4.39e−04	
Order:												
o__Alteromonadales	0	5	0	0.6	1.2	0	1	0	0.1	0.3	0.036	
o__Anaeroplasmatales	0	42	2	8.5	11.4	0	4	0	1.0	1.4	3.96e−04	
o__Bacillales	0	3	0	0.3	0.8	0	4	1	1.3	1.3	8.15e−03	
o__Bifidobacteriales	0	1	0	0.03	0.2	11	122	35	47.3	33.8	1.62e−05	
o__Enterobacteriales	0	3	0	0.4	0.8	0	1	0	0.06	0.2	7.99e−03	
o__Erysipelotrichales	1	61	12.5	16.7	14.9	10	214	80	86.2	56.9	7.50e−05	
o__Flavobacteriales	0	27	2.5	4.6	6.4	0	2	0	0.3	0.6	2.96e−04	
o__RF32	0	20	5.5	7.3	5.6	0	0	0	0	0	3.90e−09	
o__Rhodobacterales	0	22	2.5	4.4	5.2	0	3	0.5	0.7	0.9	1.95e−04	
o__Rickettsiales	0	9	2	2.4	2.2	0	9	1.5	2.3	2.4	0.021	
o__Turicibacterales	0	5	0	0.3	0.9	0	7	2	2.6	2.2	3.43e−04	
o__Verrucomicrobiales	10	4,027	195	484	747	0	3	0	0.7	0.9	4.39e−04	
o__YS2	1	59	14.5	20.6	17.9	0	8	3	2.9	2.3	1.08e−06	
Unclassified c__Alphaproteobacteria	0	2	0	0.2	0.5	0	30	4.5	6.6	7.5	2.38e−03	
Unclassified c__Bacilli	0	9	0	0.4	1.7	3	14	5.5	6.8	3.2	7.40e−08	
Family:												
f__Alteromonadaceae	0	3	0	0.3	0.7	0	0	0	0	0	0.027	
f__Anaeroplasmataceae	0	42	2	8.5	11.4	0	4	0	1.0	1.4	3.96e−04	
f__Bacillaceae	0	1	0	0.1	0.2	0	3	1	1.1	1.2	2.02e−03	
f__Bacteroidaceae	0	67	12.5	16.5	18.3	95	462	175	201	90.5	1.09e−07	
f__Bifidobacteriaceae	0	1	0	0.03	0.2	11	122	35	47.3	33.8	1.62e−05	
f__Clostridiaceae	0	59	8.5	14.6	14.7	1	609	170	207	203	8.55e−04	
f__Cryomorphaceae	0	2	0	0.3	0.5	0	0	0	0	0	4.95e−03	
f__Enterobacteriaceae	0	3	0	0.4	0.8	0	1	0	0.1	0.2	7.99e−03	
f__Erysipelotrichaceae	1	61	12.5	16.7	14.9	10	214	80	86.2	56.9	7.50e−05	
f__Flavobacteriaceae	0	27	2	4.0	5.9	0	2	0	0.2	0.5	4.69e−04	
f__Halomonadaceae	0	4	0	0.5	1	0	1	0	0.1	0.3	0.039	
f__ [Odoribacteraceae]	0	3	0	0.1	0.5	58	385	147	171	86.4	1.93e−07	
f__ [Paraprevotellaceae]	0	1	0	0.1	0.3	0	2,080	582	657	491	2.75e−05	
f__Pelagibacteraceae	0	9	2	2.3	2.2	0	5	1	1.2	1.3	0.027	
f__Peptostreptococcaceae	0	1	0	0.03	0.2	0	61	0	12.1	20.4	0.022	
f__Porphyromonadaceae	5	220	35	52.3	46.2	45	338	105	113	62.4	1.26e−03	
f__Rhodobacteraceae	0	22	2.5	4.3	5.2	0	3	0.5	0.7	0.9	2.11e−04	
f__Rikenellaceae	45	1,125	222	254	186	93	350	131	149	60.6	3.31e−03	
f__S24-7	4,965	1.3e4	1.1e4	1.1e4	1,607	8,390	1.1e4	9,994	9,831	989	0.036	
f__Staphylococcaceae	0	3	0	0.3	0.7	0	0	0	0	0	0.048	
f__Turicibacteraceae	0	5	0	0.3	0.9	0	7	2	2.6	2.2	3.43e−04	
f__Verrucomicrobiaceae	10	4,027	195	484	747	0	3	0	0.7	0.9	4.39e−04	
Unclassified o__Bacillales	0	0	0	0	0	0	1	0	0.2	0.4	0.042	
Unclassified o__Bacteroidales	0	25	9.5	10.6	5.9	3	18	9.5	10.3	4.7	9.78e−08	
Unclassified o__Lactobacillales	0	7	1	1.3	1.5	0	13	7	6.2	3.7	3.05e−05	
Unclassified o__RF32	0	20	5.5	7.3	5.6	0	0	0	0	0	3.90e−09	
Unclassified o__YS2	1	59	14.5	20.6	17.9	0	8	3	2.9	2.3	1.08e−06	
Genus:												
g__AF12	0	2	0	0.3	0.6	6	54	18	19.2	11.1	1.50e−06	
g__Akkermansia	10	4,027	195	484	747	0	3	0	0.7	0.9	4.42e−06	
g__Allobaculum	0	4	0	0.1	0.7	4	155	74	75.7	47.9	3.82e−06	
g__Anaeroplasma	0	42	2	8.5	11.4	0	4	0	1.0	1.4	3.96e−04	
g__Bacteroides	0	66	12	16.2	18.1	95	462	174	200	90.5	1.07e−07	
g__Bifidobacterium	0	1	0	0.03	0.2	11	120	34	46.6	33.2	1.57e−05	
g__Bilophila	0	0	0	0	0	0	19	1	2.7	4.7	0.024	
g__Candidatus Arthromitus	0	53	7	11.9	13.7	0	11	0	1.5	6.6	9.77e−05	
g__Clostridium	0	10	1	2.2	2.8	0	38	5.5	8.8	10.1	0.014	
g__Desulfovibrio	13	301	77.5	114	89.1	0	204	37.5	63.0	59.6	0.017	
g__ [Eubacterium]	0	2	0	0.3	0.5	0	0	0	0	0	2.57e−03	
g__Fluviicola	0	1	0	0.1	0.3	0	0	0	0	0	0.044	
g__Lawsonia	0	6	1.5	1.6	1.4	0	1	0	0.1	0.3	1.36e−07	
g__Odoribacter	0	3	0	0.2	0.5	58	385	147	171	86.4	1.93e−07	
g__Parabacteroides	5	220	35	52.3	46.2	45	338	105	112	62.4	1.26e−03	
g__Polaribacter	0	8	0	1.1	1.9	0	0	0	0	0	1.73e−03	
g__ [Prevotella]	0	1	0	0.1	0.3	0	2,077	582	657	491	2.73e−05	
g__Rikenella	0	0	0	0	0	0	39	10	11.2	12.1	1.14e−03	
g__ [Ruminococcus]	17	545	78.5	134	151	16	113	49	49.8	29.6	2.62e−03	
g__Staphylococcus	0	3	0	0.3	0.7	0	0	0	0	0	0.048	
g__Turicibacter	0	5	0	0.3	0.9	0	7	2	2.6	2.2	3.43e−04	
g__Ulvibacter	0	4	0	0.7	1.0	0	1	0	0.1	0.2	1.72e−03	
Unclassified f__Bacillaceae	0	1	0	0.1	0.2	0	3	1	1.1	1.2	2.02e−03	
Unclassified f__Bacteroidaceae	0	1	0	0.2	0.4	0	0	0	0	0	6.30e−03	
Unclassified f__Bifidobacteriaceae	0	0	0	0	0	0	3	0	0.8	1.1	8.75e−03	
Unclassified f__Clostridiaceae	0	3	0	0.6	0.8	1	589	164	194	191	4.86e−04	
Unclassified f__Coriobacteriaceae	0	7	2	2.0	1.9	0	2	1	0.7	0.8	7.48e−04	
Unclassified f__Desulfovibrionaceae	0	0	0	0	0	0	154	2.5	22.9	40.4	0.028	
Unclassified f__Enterobacteriaceae	0	2	0	0.4	0.7	0	0	0	0	0	1.73e−03	
Unclassified f__Flavobacteriaceae	0	10	1	1.5	2.3	0	1	0	0.1	0.3	8.03e−04	
Unclassified f__Pelagibacteraceae	0	9	2	2.3	2.2	0	5	1	1.2	1.3	0.027	
Unclassified f__Peptostreptococcaceae	0	1	0	0.03	0.2	0	61	0	12.1	20.4	0.022	
Unclassified f__Rhodobacteraceae	0	11	1	1.7	2.4	0	3	0	0.4	0.9	1.51e−04	
Unclassified f__Rikenellaceae	45	1,123	221	252	185	72	260	102	117	47.8	1.70e−04	
Unclassified f__S24-7	4,965	1.3e4	1.1e4	1.1e4	1,608	8,390	1.1e4	9,317	9,832	989	0.036	

WT (BZ) and chemerin KO mice have differences in Desulfovibrionaceae, Rhodobacteraceae, and Rikenellaceae abundances

To determine the impact of a loss of chemerin expression on the mouse gut microbiome, we compared the taxonomic profiles between WT (BZ) and chemerin KO mice, which were both housed at VAPAHCS. Principal coordinate analysis of both unweighted (PC1 6.65%, PC2 5.46%, PC3 5.18%) and weighted UniFrac revealed a lack of distinct separation between WT (BZ) and chemerin KO mice (Adonis: p = 0.33; Fig. 3A). However, weighted UniFrac explained more of the separation between WT (BZ) and chemerin KO mice (PC1 35.91%, PC2 15.39%, PC3 10.68%) suggesting that differences between the two genotypes are related to changes in the relative abundance of OTUs versus their presence or absence. Further analysis of differences in the abundance of different taxonomic ranks revealed a very modest number of changes between WT (BZ) and chemerin KO mice (3 significant differences at the genus level). The majority of these comprised a very minor percentage of total sequences sampled and are shown in Table 2. The largest differences between WT (BZ) and chemerin KO mice comprised ∼0.02–1.5% of total sequences and are shown in Fig. 3B. These include an increase in the abundance of Desulfovibrionaceae and unclassified members of the Rhodobacteraceae family, and a decrease in the abundance of Rikenella, in chemerin KO mice compared to WT (BZ). Considered together, this data indicates that the microbiome, in terms of relative community abundance, is not substantially altered between healthy WT (BZ) and chemerin KO mice at 6 weeks of age.

Figure 3 Wildtype and chemerin KO mice exhibit similarities in microbiome composition.

(A) Principal coordinate analysis (PCoA) of 16S sequences sampled from WT (BZ) and chemerin KO mice using weighted UniFrac. Each point represents a different sample. (B) Changes in the abundance of different taxonomic ranks between wildtype and chemerin KO mice. Families or genera that had significant differences in abundance between the two genotypes are shown. “Unclassified” denotes OTUs that have not been assigned to a specific genus but are known to be within the family specified.

Table 2 Significant differences in the abundance of bacteria between wildtype mice (BZ) and chemerin KO mice.

Statistical analysis of 16S sequences isolated from the stool of wildtype (BZ) and chemerin KO mice was performed. Significant differences at each taxonomic rank are indicated. Abundance values are shown for both genotypes and represent the number of sequences identified out of a total of 15,000 sequences analyzed per sample. “Unclassified” denotes OTUs that do not have a specific classification but are known to be within the level specified. For simplicity, unclassified ranks are removed from further levels in the table. Taxonomic ranks are represented as k, kingdom; p, phylum; c, class; o, order; f, family; g, genus; and s, species. The minimum (min), maximum (max), median (med), and mean number of observed sequences in a single sample are shown. SD, standard deviation.

	Wildtype (BZ)	Chemerin KO		
Feature	Min	Max	Med	Mean	SD	Min	Max	Med	Mean	SD	p-value	
Phylum:												
p__Cyanobacteria	0	8	3.5	3.2	2.4	0	4	1	1.4	1.3	0.01	
p__Proteobacteria	20	214	102	97.8	58.5	30	402	91	164	109	0.03	
Class:												
c__4C0d-2	0	8	3	2.9	2.3	0	4	1	1.3	1.2	0.015	
c__Deltaproteobacteria	19	212	83	88.8	59.9	22	392	74	152	112	0.043	
c__Flavobacteriia	0	2	0	0.3	0.6	0	12	0	2.4	3.3	0.016	
Order:												
o__Desulfovibrionales	19	212	83	88.7	60.0	22	392	74	152	112	0.044	
o__Flavobacteriales	0	2	0	0.3	0.6	0	12	0	2.4	3.3	0.016	
o__Rhodobacterales	0	3	0.5	0.7	0.9	0	22	1	4.4	6.5	0.028	
o__YS2	0	8	3	2.9	2.3	0	4	1	1.3	1.2	0.015	
Family:												
f__Cryomorphaceae	0	0	0	0	0	0	1	0	0.2	0.4	0.042	
f__Desulfovibrionaceae	19	212	83	88.7	60.0	22	392	74	152	112	0.044	
f__Flavobacteriaceae	0	2	0	0.2	0.6	0	11	0	1.8	3.1	0.041	
f__Rhodobacteraceae	0	3	0.5	0.7	0.9	0	22	1	4.4	6.5	0.028	
Unclassified o__YS2	0	8	3	2.9	2.3	0	4	1	1.3	1.2	0.015	
Genus:												
g__Lactococcus	0	3	0	0.5	0.9	0	0	0	0.00	0.00	0.035	
g__Rikenella	0	39	10	11.2	12.1	0	24	1	3.7	6.2	0.028	
Unclassified f__Rhodobacteraceae	0	3	0	0.7	0.9	0	20	1	4.3	6.1	0.024	

Figure 4 Wildtype and CMKLR1 KO mice exhibit a modest separation in bacterial diversity.

Principal coordinate analysis (PCoA) of 16S sequences sampled from WT (CS) and CMKLR1 KO mice using weighted UniFrac. Each point represents a different sample. (B) Changes in the abundance of different bacteria at various taxonomic ranks between wildtype and CMKLR1 knockout mice. Bacterial ranks that composed more than 1% of sequences sampled and had significant differences in abundance between the two genotypes are shown. “Unclassified” denotes OTUs that have not been assigned to a specific genus but are known to be within the family specified.

CMKLR1 KO mice have an increased abundance of Akkermansia and Prevotella compared to WT (CS)

We next examined differences in gut microbiome profiles between WT (CS) and CKMLR1 KO mice housed at Dalhousie University. Notably, there was no significant difference in body weight at the times of fecal sample collection between wildtype (6 weeks, 18.0 ± 0.4 g; 8 weeks 19.1 ± 0.5 g) and CMKLR1 KO (6 weeks, 17.1 ± 0.4 g; 8 weeks 18.2 ± 0.5 g) mice. Principal coordinate analysis of the unweighted UniFrac revealed no obvious pattern of separation between wildtype and CMKLR1 KO mice, and only explained a relatively minor proportion of the separation between samples (PC1 5.11%, PC2 4.48%, PC3 3.00%). However, principal coordinate analysis of the weighted UniFrac (Fig. 4A) explained a higher amount of variability between samples (PC1 37.64%, PC2 21.94%, PC3 6.95%). There was a trend for a left-shift in CMKLR1 KO mice on PC3, suggesting that PC3 explains a small amount of separation between genotype, although not statistically significant (Adonis: p = 0.1). Further analysis of OTU abundances revealed a number of significant differences between WT (CS) and CMKLR1 KO mice (Table 3). Importantly, the phylum Firmicutes made up a considerable proportion of all sequences in WT mice (∼18%) and was significantly higher in CMKLR1 KO mice (∼20%; Fig. 4B). However, it is unclear which bacteria make up this difference as there were no differences in Firmicutes at lower taxa levels. The majority of other significant differences occurred at lower levels, including 8 different taxa at the genus level (Table 3). A number of these differences made up a relatively minor proportion of total sequences (less than 0.2% of total sequences), although there were several differences that comprised >1% of total sequences. These include Akkermansia, which comprised ∼3-5% of the microbiome in WT animals but only ∼1% in CMKLR1 KO animals (Fig. 4B). Additionally, Prevotella (note this is a different genus from [Prevotella] discussed earlier with reference to Fig. 2B, where both are classified in the order Bacteroidales, but Prevotella is a member of the family Prevotellaceae, whereas [Prevotella] is a member of the family [Paraprevotellaceae]) and unclassified members of the Rikenellaceae family decreased ∼1.5-fold in CMKLR1 KO animals compared to WT (CS) (Fig. 4B).

Table 3 Significant differences in the abundance of bacteria between wildtype mice (CS) and CMKLR1 KO mice.

Statistical analysis of 16S sequences isolated from wildtype (CS) and CMKLR1 KO mice was performed. Significant differences at each taxonomic rank are indicated. Abundance values are shown for both genotypes and represent the number of sequences identified out of a total of 15,000 sequences analyzed per sample. “Unclassified” denotes OTUs that do not have a specific classification but are known to be within the level specified. For simplicity, unclassified ranks are removed from further levels in the table. Taxonomic ranks are represented as k, kingdom; p, phylum; c, class; o, order; f, family; g, genus; and s, species. The minimum (min), maximum (max), median (med), and mean number of observed sequences in a single sample are shown. SD, standard deviation.

	Wildtype (CS)	CMKLR1 (KO)		
Feature	Min	Max	Med	Mean	SD	Min	Max	Med	Mean	SD	p-value	
Phylum:												
p__Cyanobacteria	1	59	15.5	20.9	17.9	1	54	7	12.7	12.6	0.025	
p__Firmicutes	1,052	5,147	245	274	1,093	868	5,614	324	334	115	0.022	
p__Verrucomicrobia	10	4,027	195	484	747	0	774	41	148	229	0.013	
Unclassified k__Bacteria	0	40	2	5	7.9	0	12	1	2	2.9	0.035	
Unclassified unassigned	97	483	200	213	69.3	75	318	175	183	49.0	0.031	
Class:												
c___4C0d-2	1	59	14.5	20.6	17.9	1	54	7	12.5	12.6	0.027	
c___Verrucomicrobiae	10	4,027	195	484	747	0	774	41	148	229	0.013	
Order:												
o__Anaeroplasmatales	0	42	2	8.5	11.3	0	45	1	3.4	7.4	0.026	
o__Thiotrichales	0	0	0	0	0	0	1	0	0.1	0.3	0.023	
o__Verrucomicrobiales	10	4,027	195	484	747	0	774	41	148	229	0.013	
o__YS2	1	59	14.5	20.6	17.9	1	54	7	12.5	12.6	0.027	
Family:												
f__Anaeroplasmataceae	0	42	2	8.5	11.4	0	45	1	3.4	7.4	0.026	
f__Bacteroidaceae	0	67	12.5	16.5	18.3	0	42	6	9	10.7	0.036	
f__Piscirickettsiaceae	0	0	0	0.00	0.00	0	1	0	0.1	0.3	0.023	
f__Prevotellaceae	18	995	376	436	254	10	1,050	224	303	252	0.026	
f__Rikenellaceae	45	1,125	222	254	186	35	485	171	181	92.2	0.039	
f__Verrucomicrobiaceae	10	4,027	195	484	747	0	774	41	148	229	0.013	
Unclassified o__Bacteroidales	0	25	9.5	10.6	6	1	20	4.5	5.6	4.4	9.33e−05	
Unclassified o__YS2	1	59	14.5	20.6	17.9	1	54	7	12.5	12.6	0.027	
Genus:												
g__Akkermansia	10	4,027	195	484	747	0	774	40.5	148	229	0.013	
g__Anaeroplasma	0	42	2	8.47	11.3	0	45	1	3.38	7.4	0.026	
g__Bacteroides	0	66	12	16.2	18.1	0	42	6	8.9	10.7	0.039	
g__Coprococcus	0	121	14	26.3	29.4	1	35	10	11.8	7.62	6.67e−03	
g__Prevotella	18	955	376	436	254	10	1,050	224	303	252	0.026	
Unclassified f__Peptococcaceae	0	3	0	0.5	0.8	0	4	1	1.1	1.2	0.014	
Unclassified f__Piscirickettsiaceae	0	0	0	0	0	0	1	0	0.1	0.3	0.023	
Unclassified f__Rikenellaceae	45	1,125	222	254	185	35	485	171	181	92.2	0.039	

Akkermansia and Prevotella abundance are negatively associated with body weight and exhibit similar patterning to WT (CS) and CMKLR1 KO mice

Previous studies have established that Akkermansia abundance is negatively correlated with body weight and glucose tolerance in rodent and humans (Everard et al., 2011; Qin et al., 2012; Santacruz et al., 2010). Studies have also shown that a high abundance of Prevotella is correlated with obesity and impaired glucose tolerance (Ellekilde et al., 2014; Okeke, Roland & Mullin, 2014). Consistent with this, we observed a negative correlation between the weight of each mouse at the time of stool collection (6 or 8 weeks) and Akkermansia (R =  − 0.361) or Prevotella (R =  − 0.395; Fig. 5) abundance within each stool sample. However, this was significant only for CMKLR1 KO mice, but not WT (CS).

Figure 5 Changes in Akkermansia and Prevotella abundance are negatively correlated with total body mass in CMKLR1 KO mice.

The body weight of each WT (CS; A and C) or CMKLR1 KO (B and D) mouse at the time of fecal sample collection was plotted against the abundance of Akkermansia (A and B) or Prevotella (C and D) within each sample.

Finally, as significant differences in abundance of Akkermansia and Prevotella were observed in samples from WT (CS) and CMKLR1 KO mice, we investigated whether differences in these genera explained the separation between genotypes observed earlier. We categorized samples as having a low or high abundance of Akkermansia or Prevotella based on the average sequence count. A high abundance of either Akkermansia (Adonis: p = 0.007) or Prevotella (Adonis: p = 0.002) separated the samples towards the right on PC3 (Fig. 6). Interestingly, this pattern is similar to that seen between WT (CS) and CMKLR1 KO animals (Fig. 4A).

Figure 6 Separation along the PC3 axis is explained by differences in Akkermansia and Prevotella abundance.

Principal coordinate analysis (PCoA) of 16S sequences sampled from WT (CS) and CMKLR1 KO mice using weighted UniFrac. Each point represents a different sample. Samples with a high or low abundance of Akkermansia (A) or Prevotella (B) are indicated with different colors.

Discussion

Over the past 10–15 years, convergent lines of investigation have implicated both gut dysbiosis and chemerin signaling with the development of metabolic and inflammatory disorders. In this study, we investigated the relationship between the two by comparing the diversity and composition of the gut microbiome in fecal samples obtained from mice with normal or reduced levels of chemerin/CMKLR1 signaling. These findings provide an essential assessment of the microbiome profile in healthy chemerin and CMKLR1 KO mice. We discovered several key differences in microbiota composition that have implications for the study of metabolic and inflammatory diseases discussed in further detail below.

The environment is a significant factor in the composition of gut microbiota

The most striking result from this study was a dramatic difference in gut microbiome composition between Dalhousie University and VAPAHCS. The animals used in this experiment were identical in terms of background strain, vendor source of C57/Bl6 mice, age, sex, breeding strategy, and pattern of co-housing. In addition, DNA isolation from fecal samples and subsequent 16S rRNA sequencing was performed for samples obtained from both facilities at the same time, eliminating any potential differences in sample preparation or sequencing bias. The only notable differences in study design included the type of rodent chow used and the physical location where the mice were housed. However, despite attempts to keep study conditions as similar as possible, there was a trend for increased species richness at VAPAHCS and a large number of differences in bacterial abundance identified between the two facilities. Both unweighted and weighted UniFrac metrics explained a relatively high percentage of variation between the two facilities, indicating that the differences in microbiota composition were due to changes in the presence of particular bacterial species as well as the abundance of common species. These include increased levels of Bifidobacterium, Clostridium, Bacteroides, Prevotella, Odoribacter, and decreased abundance of Akkermansia, Ruminococcus, and Desulfovibrio in VAPAHC compared to Dalhousie University. It is unclear what factors are responsible for the differences in microbiome profile between facilities. The rodent chow used at VAPAHCS excludes alfalfa, which may explain a decrease in photosynthesizing Cyanobacteria compared to mice at Dalhousie University. Other dietary factors, such as the percentage of calories from fat (18% versus 14%) might influence the composition of other gut bacteria. Additionally, differences in the cleanliness of the rooms or exposure to pathogens might influence microbial composition. Further identification of the environmental factors that influence microbiome composition will be essential in order to increase reproducibility and recognize potential confounding factors in future studies.

These results are consistent with previous studies that have demonstrated that the microbiome of mice used in biomedical research is greatly influenced by environment. For example, studies have shown that genetically similar mice obtained from different commercial vendors have profound differences in the richness and diversity of fecal microbial populations (Ericsson et al., 2015). Consistent with this, environmental reprogramming of microbiota can ameliorate or accelerate the development of diseases such as obesity and T2D (Ussar et al., 2015). Notably, the abundance of several genera that were identified to be significantly different between Dalhousie University and VAPAHCS (e.g., Bifidobacterium, Clostridium, Bacteroides, Prevotella, and Akkermansia) have been associated with changes in adiposity, T2D, and/or IBD (Barlow, Yu & Mathur, 2015; Keeney et al., 2014; Zhang et al., 2015). This might have important implications for studies that investigate the role of chemerin signaling in vivo. For instance, multiple groups have presented conflicting results on the adipose and glucose phenotype of CMKLR1 KO mice. For example, previously we reported that CMKLR1 KO mice have reduced adiposity but worsened glucose tolerance when challenged with a high fat diet (Ernst et al., 2012). In contrast, other independent research groups published findings indicating that CMKLR1 KO mice are more susceptible to weight gain (Rouger et al., 2013; Wargent et al., 2015). However, these studies also provided disparate findings with one reporting worsened, and the other reporting unchanged, glucose tolerance. A further study reported no impact of CMKLR1 loss on either weight gain or glucose tolerance (Gruben et al., 2014). Interestingly, the source, and presumably genetics, of the transgenic mouse line used is identical in three of these studies (Ernst et al., 2012; Rouger et al., 2013; Wargent et al., 2015), making the reasons for such discrepancies in the phenotype of CMKLR1 KO mice unclear. Given the profound effect of the environment on microbiome composition, and the known influence of the microbiome on the pathogenesis of disease, it is possible that differences in microbiome composition confounded the results of these studies. It would be valuable to collect stool samples from CMKLR1 KO mice from each of the different facilities used and compare the microbiome composition between the different mouse colonies. Of note, it would also be interesting to compare levels of total and bioactive circulating chemerin in mouse colonies from different animal facilities. Together, this information would be useful not only for studies involving obesity and T2D, but also other diseases that are known to be influenced by the microbiome.

Changes in microbiome composition with a loss of chemerin signaling

As a result of the differences in microbiome composition between Dalhousie University and VAPAHCS, we were limited to comparing chemerin and CMKLR1 KO mice to their respective WT mice within each facility. For both genotypes, there were no changes in alpha diversity compared to wildtype. In contrast to the comparison between the two facilities, there was very little variability between WT and KO mice explained using unweighted UniFrac metrics. An increased amount of variability was explained by the weighted UniFrac, suggesting that differences in microbiome composition between both chemerin and CMKLR1 KO mice compared to their respective WT mice are due to differences in the abundance of common taxa. Within VAPAHCS, there were very few changes in microbiome composition between chemerin KO and WT mice. These included changes in Rikenella, Lactococcus, and members of the Desulfovibrionaceae and Rhodobacteraceae families, which have been linked with both metabolic and inflammatory diseases (Keeney et al., 2014; Okeke, Roland & Mullin, 2014). There were significantly more differences in the abundance of microbiota identified between WT and CMKLR1 KO mice at Dalhousie University, including a relatively high number of differences at the genus level. The largest differences in the abundance of bacteria were in Akkermansia and Prevotella, which together comprised ∼3–8% of the total bacteria isolated from the gut of WT mice and explained ∼6% of variability between WT and CMKLR1 KO mice.

It was not possible to directly compare differences between CMKLR1 and chemerin KO mice because of the differences in microbiome composition observed between facilities. However, both chemerin and CMKLR1 KO mice exhibited decreases in the abundance of Rikenella compared to WT. A previous study identified that Rikenella was 1 of 10 genera to be affected by genotype in the feces of obese diabetic db/db mice compared to lean controls (Geurts et al., 2011). This suggests a potential link between chemerin and metabolic disease. It was surprising that there were more differences in bacterial abundance identified between CMKLR1 KO and WT mice compared to chemerin KO mice. It is possible that this is an artifact from the differences in environment. For example, Akkermansia was nearly undetectable in mice housed at VAPAHCS, so it is difficult to predict whether the abundance of Akkermansia would be altered in chemerin KO mice in a manner similar to CMKLR1 KO mice if the genus were present in the bacterial community. Alternatively, ligands other than chemerin, including resolvin E1 and beta-amyloid, have been reported to act as agonists at CMKLR1, although these remain to be confirmed by independent groups (Arita et al., 2005; Peng et al., 2015). As such, it is possible that chemerin-independent CMKLR1 signaling contributed to differences in the abundance of microbial populations between chemerin and CMKLR1 KO mice. Future studies that house all groups of mice in the same facility will enable the direct comparison of microbiome profiles between chemerin, CMKLR1, and WT mice.

Relationship between the microbiome and CMKLR1 signaling in adiposity

Chemerin signaling has been positively associated with obesity and T2D. In addition to the increased Rikenella abundance in chemerin and CMKLR1 KO mice discussed above, the abundance of several bacteria that are known to correlate with adiposity and glucose tolerance were altered between wildtype and CMKLR1 KO mice. These include Akkermansia, Bacteroides, and Prevotella. Recent studies have highlighted a direct role for Akkermansia, a mucin-degrading bacterium, in obesity, where treatment of mice with Akkermansia has been shown to reduce high fat diet (HFD)-induced metabolic disorders including fat mass gain, metabolic endotoxemia, adipose tissue inflammation, and insulin resistance (Everard et al., 2013). This demonstrates that specific bacteria that have been identified to correlate with body weight are able to exert direct metabolic effects on the host. Similar to previous reports (Everard et al., 2011; Qin et al., 2012; Santacruz et al., 2010), we identified a negative relationship between both Akkermansia and Prevotella abundance and total body weight. Interestingly, this relationship was only significant for CMKLR1 KO mice. A higher abundance of Akkermansia and Prevotella populations over a small weight range might explain the lack of significant correlation in WT mice. Alternatively, the influence of a loss of CMKLR1 on the abundance of these genera, or the effect of these genera on the body weight of CMKLR1 KO mice, might be differentially regulated. A differential effect of Akkermansia or Prevotella populations on CMKLR1 KO mice would help to explain previously-reported differences in adiposity and glucose tolerance in the absence of CMKLR1 signaling (Ernst et al., 2012; Rouger et al., 2013; Wargent et al., 2015). Of note, it would be interesting to analyze the correlation between fecal microbiota abundance and adipokine levels associated with adiposity such as leptin and adiponectin. Future studies that directly address the role of Akkermansia and Prevotella in CMKLR1 KO mice, including more time points to observe mice over a longer period of time, DEXA analysis for body composition, and the effect of a high-fat diet (HFD), will prove useful in further examining the relationship between the gut microbiome on the adiposity and glucose tolerance phenotype of CMKLR1 KO mice.

Relationship between the microbiome and CMKLR1 signaling in IBD

Previous studies have demonstrated a relationship between the abundance of a number of bacterial populations on the development of IBD (Keeney et al., 2014). In general, these studies tend to associate a pro-inflammatory microbial population with an increased risk for the development or increased severity of IBD. Increases in the abundance of Bacteroides, Bifidobacterium, Clostridium, and a decrease in Akkermansia levels were observed in samples from VAPAHCS compared to Dalhousie University. These species have been correlated with the prevalence of IBD, suggesting that mice at the two facilities might have altered susceptibility to IBD. Consistent with this, conflicting results have been presented regarding the effect of systemic chemerin injection on the development of IBD (Dranse et al., 2015; Lin et al., 2014). Within Dalhousie University, CMKLR1 KO mice exhibited a decrease in Akkermansia and Prevotella species. There are a number of studies that associate both of these taxa with the severity of IBD. In particular, it has been suggested that changes in mucin degradation resulting from altered Akkermansia abundance influence epithelial cell layer integrity and the pathogenesis of IBD. Consistent with this, Akkermansia abundance is reduced in patients with both UC and CD (Png et al., 2010) and administration of Akkermansia has been shown to protect against the progression of DSS-induced colitis (Kang et al., 2013). In contrast, other groups have shown that Akkermansia exacerbates the severity of disease in mouse models of Salmonella typhimurium-induced gut inflammation and inflammation-associated colorectal cancer in mice (Ganesh et al., 2013; Zackular et al., 2013). We previously demonstrated that CMKLR1 KO mice develop clinical signs of DSS-induced colitis more slowly than wildtype mice (Dranse et al., 2015), which supports the latter findings. Additionally, a high abundance of Prevotella is correlated with the prevalence of UC in humans and increased epithelial inflammation in a colitis mouse model (Lucke et al., 2006; Scher et al., 2013), which is consistent with CMKLR1 KO mice exhibiting slower disease progression. However, it is important to note that we did not observe any changes in key bacteria associated with IBD such as Faecalibacterium. Based on this observation, and the correlative nature of our findings, future studies that directly examine the relationship between chemerin signaling and the microbiome on the pathogenesis of IBD are warranted.

Influence of chemerin as an antibacterial agent on the microbiome

Previous studies have demonstrated that chemerin acts as an antimicrobial agent and prevents the growth of E. coli, K. pneumoniae, and S. aureus (Banas et al., 2013; Kulig et al., 2011). In this study, we did not observe any of these species, which is not surprising as these pathogenic species would not normally be present in the healthy gastrointestinal tract. An extremely limited number of unclassified Enterobacteriaceae and Staphylococcus sequences (maximum 5 sequences out of 15,000 total), for which E. coli and S. aureus may have been classified under, were identified in samples from Dalhousie University and VAPAHCS. In the absence of these species, we are unable to determine whether the antimicrobial effects of chemerin are occurring in the mouse gut. However, due to the very low number of differences in gut microbiota between chemerin KO and WT mice, it seems unlikely that chemerin is exerting an antimicrobial effect under healthy conditions. This is consistent with the low basal expression of chemerin in the gut (Dranse et al., 2015). It seems more likely that chemerin plays an antimicrobial role to protect against pathogen invasion and exhibits increases in expression and activation in disease states when required. This concept is supported by evidence that pathogen-derived enzymes such as S. aureus-derived Staphopain B are able to elevate levels of bioactive chemerin (Kulig et al., 2007). Future studies that directly examine the susceptibility of chemerin KO animals following infection with pathogenic bacteria, compare the microbiome profiles of chemerin KO animal in challenged conditions, and investigate chemerin isoform distribution in the gastrointestinal tract following changes in microbiota composition, will provide further insight into the role of chemerin as an antibacterial agent.

Future studies

It is important to note that this study was performed using healthy, young mice; however, both CMKLR1 and chemerin KO mice exhibit differences in phenotype under stressed conditions. For example, chemerin KO mice exhibit exacerbated glucose intolerance on an HFD (Takahashi et al., 2011) and CMKLR1 KO mice exhibit differences in adiposity and glucose tolerance when fed an HFD compared to a low-fat diet (Ernst et al., 2012). Additionally, CMKLR1 KO mice develop signs of clinical illness more slowly in an experimentally-induced colitis model (Dranse et al., 2015). As the microbiome is associated with the development of these disorders, it will be interesting to examine whether differences in microbiome composition in chemerin and CMKLR1 KO mice under stressed conditions such as a HFD or chemically-induced colitis influence the development of inflammatory and metabolic diseases. Additionally, in the current study we co-housed wildtype and KO mice at weaning, and it is possible that transfer of microbes occurred between genotypes in the same cage as previously demonstrated (Yoshimura et al., 2018). Therefore, future studies that compare the microbiota composition of mice housed alone versus with a different genotype may identify further differences in microbiome profiles. Furthermore, studies that directly investigate the role of chemerin on the growth and/or survival of particular microbiota in diseased states will be valuable. For example, chemerin is secreted at nearly undetectable levels in the colon under basal conditions, but levels of chemerin secretion and activation increase dramatically when inflamed (Dranse et al., 2015). This suggests that the impact of increased chemerin secretion in the colon on the microbiome might be more important in a disease context than in normal physiology. Additionally, in this study, we investigated the microbiome in fecal samples. As changes in microbiota composition and activity have been shown to vary throughout the length of the GI tract (Gu et al., 2013), analysis of samples from the small intestines and cecum may provide further information on the gut microbiome. Finally, future studies that directly test the influence of microbial species on chemerin-related functions will be essential to determine the direct role of the microbiome in relation to chemerin signaling.

Conclusions

In conclusion, this is the first study that investigates the link between chemerin/CMKLR1 and microbiome composition. The information gained from these studies will be valuable in the development of studies that directly investigate the relationship between chemerin signaling and the microbiome in the pathogenesis of multiple disease states. Despite attempts to maintain similar experimental conditions between different facilities, these results highlight the impact of the environment on microbiome composition. Importantly, changes in bacteria that are known to influence chemerin-associated diseases were differentially present between facilities, highlighting that the microbiome might be a confounding factor when studying the role of chemerin signaling in obesity and inflammation in vivo. Additionally, we demonstrated that differences in the abundance of Akkermansia and Prevotella, both established to impact adiposity and glucose tolerance, are correlated with body weight and are decreased in CMKLR1 KO mice. This indicates that the microbiome might influence the development of metabolic and inflammatory diseases in relation to chemerin signaling. This has potential implications for the development of novel methods to modify disease risk through lifestyle changes such as dietary intervention, exercise, use of probiotics, or fecal transplantation. Future studies that house mice in the same environment, focus on other methods to modify chemerin signaling, incorporate disease models, and predict functional changes in microbial activity, will be informative to fully elucidate the relationship between chemerin signaling and the microbiome on health and disease.

Additional Information and Declarations

Competing Interests

Author Contributions

Animal Ethics

Data Availability

The authors declare thre are no competing interests.

Helen J. Dranse conceived and designed the experiments, performed the experiments, analyzed the data, prepared figures and/or tables, authored or reviewed drafts of the paper, approved the final draft.

Ashlee Zheng performed the experiments, analyzed the data, authored or reviewed drafts of the paper, approved the final draft.

André M. Comeau performed the experiments, analyzed the data, prepared figures and/or tables, authored or reviewed drafts of the paper, approved the final draft.

Morgan G.I. Langille and Brian A. Zabel conceived and designed the experiments, analyzed the data, contributed reagents/materials/analysis tools, authored or reviewed drafts of the paper, approved the final draft.

Christopher J. Sinal conceived and designed the experiments, performed the experiments, analyzed the data, contributed reagents/materials/analysis tools, prepared figures and/or tables, authored or reviewed drafts of the paper, approved the final draft.

The following information was supplied relating to ethical approvals (i.e., approving body and any reference numbers):

Animal protocols were approved by Dalhousie University Committee on Laboratory animals (14- 064) in accordance with the Canadian Council on Animal Care guidelines and the Institutional Animal Use and Care Committee at the Veterans Affairs Palo Alto Health Care System.

The following information was supplied regarding data availability:

Raw reads for this study have been submitted to the European Nucleotide Archive under accession PRJEB25165.

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
