# Peer review of "The impact of chemerin or chemokine-like receptor 1 loss on the mouse gut microbiome"

_PeerJ, doi:10.7717/peerj.5494_

## Round 0.1 · original submission · Major Revisions

Based on the valuable critique & suggestions by the three referees, the study needs improvement. Some issues need to be discussed in the rebuttal letter if further experimentation is not possible. Some data seem to be missing (e.g., weight?)

Reviewer 1 ·

Basic reporting

The manuscript: „The impact of chemerin or chemokine-like receptor 1 loss on the mouse gut microbiome“ by Dranse et al. shows a linkage between chemerin and / or CMKLR1 and microbiome composition. They further provide data on the fecal microbiome when animals were housed in different facilities.

This is a well performed study and the aim of the present investigation is clearly described.
English is fine and citted literature is appropriate.

Data are nicely described and presented (with one exception indicated below)

Experimental design

The experimental design is appropriate. Methods are well described. The authors present original research which is within the scope of the journal.

The aim of the present investigation is clearly described. Analysis of microbiome composition is supposed to help several issues relevant in health / disease. Further, this may be one reason for different findings in animal studies performed in different facilities.

Ethical approval for animal studies has been obtained. and results look valid.

Validity of the findings

First study analyzing the microbiome of chemerin / CMKLR1 KO mice. Study design is appropriate and number of animals analyzed is o.k.

Data are well discussed and respective literature is cited and discussed.

Regarding statistics it is unclear whether authors corrected for multiple comparisons,

Additional comments

The manuscript: „The impact of chemerin or chemokine-like receptor 1 loss on the mouse gut microbiome“ by Dranse et al. shows a linkage between chemerin and / or CMKLR1 and microbiome composition. They further provide data on the fecal microbiome when animals were housed in different facilities.

This is a well performed study and the aim of the present investigation is clearly described. Ethical approval for animal studies has been obtained. The experimental design is appropriate and results look valid.

One limitation of the study is that body weight of some mice was not recorded. Authors may measure serum glucose or another well described marker of body fat like adiponectin and evaluate any correlations with fecal microbiota.

It is unclear whether authors corrected for multiple comparisons.

Wildtype mice of the two facilities differ in their fecal microbiota. The authors may analyze serum chemerin to find out whether levels are comparable in both strains.

In figure 1 the graph for WT (CS) is not visible, data have to be shown in a different way.

Minor:
“IBD pathogenesis have demonstrated that local 74 chemerin expression, secretion, and activation increase locally in the colon and are positively associated with severity of inflammation (Dranse et al. 2015; Lin et al. 2014; Weigert et al. 2010).”
Locally or local may be deleted.

“Genomic DNA was isolated from fecal sample samples using the PowerFecal DNA”
please delete “sample”

In “future studies” the term LFD is used but is not explained before.

“In conclusion, this is the first study that investigates the link between chemerin signaling and microbiome composition“
The term “signaling” may be removed here and elsewhere especially regarding that chemerin knock-out mice do not display major changes in the fecal microbiome. Authors can not be sure that changes observed in CMKLR1 KO are related to chemerin.

·

Basic reporting

In the present study, the authors assessed the microbiome composition of fecal samples isolated from WT, chemerin, or CMKLR1 KO mice using Illumina-based sequencing. Chemerin KO and CMKLR1 KO mice were housed at two different facilities and WT mice housed at each facility were used as control. There was a dramatic difference in microbiome composition between two facilities. There were only minor differences in bacterial abundance between WT and chemerin KO mice, but significantly more differences in taxa abundance between WT and CMKLR1 KO mice. Specifically, CMKLR1 KO mice exhibited decreased abundance of Akkermansia and Prevotella, which correlated with body weight in CMKLR1 KO, but not WT. This study is incomplete because the authors analyzed the microbiome composition only after cohousing. The manuscript is clearly written.

Experimental design

1. KO mice were cohoused with WT mice for three weeks before collection of fecal pellets. What was the reason. As this reviewer recently reported (J. Immunol. 2018, 200:2174-2185), the composition of microbiota can be markedly altered by cohousing.

2. Fecal pellets were collected from female mice. What was the reason? For example, male mice were previously used in their colitis study.

3. Line 143-144: Mice were weighed. What was the result? Did the authors still detect a difference in the weight between WT and KO mice after cohousing?

4. Line 384-386: It would be valuable to collect----.” This experiment should be done.

5. Line 508-509: The authors cite their previous study and describe that CMKLR1 KO mice develop signs of clinical illness more slowly in an experimentally-induced colitis model (Dranse et al. 2015). In the study, littermates were used, which may have contributed to the very marginal difference between WT and KO.

Validity of the findings

Data may be clear but the study is incomplete.

Reviewer 3 ·

Basic reporting

The work entitled "The Impact of Chemerin or Chemokine-Like Receptor Loss on the Mouse Gut Microbiome" by HJ Dransen and collaborator aimed to detect changes in the microbiota in wild type mice compared to know-out mice impaired for the Chemerin, an adipocyte derived signaling molecule that serves as a ligand activator of Chemokine-like receptor 1(CMKLR1) and the proper CMKLR1. This pathways are responsible for the regulation of metabolism and inflammation processes.

The literature, the structure of the work, the English form, proposed tables and figures are appropriate.

All figures included in the PDF are of very low quality and must be improved.

Experimental design

In general the applied strategy based on 16S rDNA amplicon metataxonomy is acceptable, the authors highlighted difference in gut microbiome by the means of fecal pellet DNA extraction and 16S rDNA amplicon Illumina sequencing, between two groups of KO mice for Chemerin and CMKLR1 from two different laboratory respectively.

Bioinformatics and statistical approaches are properly applied.

Validity of the findings

Main differences have been observed between the mice colonies belonging to the two laboratories participating to the study. Within same facility, WT and KO mice showed very small difference mainly related to few groups of bacteria. Such as Akkermansia and Prevotella, both negatively correlate to body mass as also described in other studies. Although this seems to add few knowledge to the field due to the high dissimilarity in microbiome found between the two laboratories, the author succeeded to reduce the confounding factor of the high microbiome diversity between KO and wild type mice from each laboratory, identifying a correlation with genera (Akkermansia and Prevotella), well known to interact in several gut-related pathologies. In such a way, they demonstrated correlation between these genera and the knocking out of adipocyte signaling pathway.

Additional comments

Minor changes
Just at the beginning of the introduction, Microbiome is described as commensal relationship. This is surely correct but is restrictive. It is widely demonstrated that on top of all possible interactions, the great niche landscape offered by the gastrointestinal tract give space also, for example, to mutualism, strict dependencies, probably parassitism, etc. Please extend a little bit more this state.

Lines 49-51. This number have been recently revised using modern data and bibliographic revision in "Are We Really Vastly Outnumbered? Revisiting the Ratio of Bacterial to Host Cells in Humans" (Cell. Volume 164, Issue 3, 28 January 2016, Pages 337-340), please add this reference.
Lines 65-67, the references cited in these lines are not consistently proposed with the format "(...see (Rourke et al. 2013))." The nested parenthesis could be removed.
68-69, same as previous point.

---

## Round 0.2 · accepted · Accept

Apologies for all the delay this manuscript has witnessed. The reviewers have no further concerns.

# Reviewer 1 ·

Basic reporting

The manuscript: „The impact of chemerin or chemokine-like receptor 1 loss on the mouse gut microbiome“ by Dranse et al. shows a linkage between chemerin and / or CMKLR1 and microbiome composition. They further provide data on the fecal microbiome when animals were housed in different facilities.

This is a well performed study and the aim of the present investigation is clearly described.
English is fine and cited literature is appropriate.

Data are nicely described and presented

Experimental design

The experimental design is appropriate. Methods are well described. The authors present original research which is within the scope of the journal.

The aim of the present investigation is clearly described. Microbiome composition is supposed to be relevant in various diseases. Further, findings presented here may explain variations in animal studies performed in different facilities.

Ethical approval for animal studies has been obtained and results look valid.

Validity of the findings

First study analyzing the microbiome of chemerin / CMKLR1 KO mice. Study design is appropriate and number of animals analyzed is o.k.

Data are well discussed and respective literature is cited and discussed.

Additional comments

Authors have addressed all my previous comments

·

Basic reporting

No comment

Experimental design

No comment

Validity of the findings

No comment

Additional comments

No comment

Reviewer 3 ·

Basic reporting

The authors fulfilled comments and improved the manuscript accordingly.

Experimental design

The authors fulfilled comments and improved the manuscript accordingly.

Validity of the findings

The authors fulfilled comments and improved the manuscript accordingly.